# OpenAD: Open-World Autonomous Driving Benchmark for 3D Object Detection

**Zhongyu Xia**[1]    **Jishuo Li**[1]    **Zhiwei Lin**[1]    **Xinhao Wang**[1]
**Yongtao Wang**[1✉]    **Ming-Hsuan Yang**[2]

[1]Wangxuan Institute of Computer Technology, Peking University
[2]University of California, Merced

{xiazhongyu,zwlin,wangxinhao,wyt}@pku.edu.cn
lijishuo@stu.pku.edu.cn    mhyang@ucmerced.edu

## Abstract

Open-world perception aims to develop a model adaptable to novel domains and various sensor configurations and can understand uncommon objects and corner cases. However, current research lacks sufficiently comprehensive open-world 3D perception benchmarks and robust generalizable methodologies. This paper introduces OpenAD, the first real open-world autonomous driving benchmark for 3D object detection. OpenAD is built upon a corner case discovery and annotation pipeline that integrates with a multimodal large language model (MLLM). The proposed pipeline annotates corner case objects in a unified format for five autonomous driving perception datasets with 2000 scenarios. In addition, we devise evaluation methodologies and evaluate various open-world and specialized 2D and 3D models. Moreover, we propose a vision-centric 3D open-world object detection baseline and further introduce an ensemble method by fusing general and specialized models to address the issue of lower precision in existing open-world methods for the OpenAD benchmark. We host an online challenge on EvalAI (2D & 3D). Data, toolkit codes, and evaluation codes are available at https://github.com/VDIGPKU/OpenAD.

## 1 Introduction

3D perception, which conforms to practical spatial definitions and physical laws, has become indispensable in autonomous driving systems. In the pursuit of advanced autonomous driving, the necessity for open-world capabilities has been recognized. The two most pivotal factors in open-world perception are domain generalization and being open ended. **Domain generalization** refers to the performance of a model when faced with new streets, regions or countries, as well as varied vehicle types. Within 3D perception for autonomous driving, existing methodologies [28, 1] for evaluating scenario generalization entail training on a specific dataset and then transferring the trained model to a distinct dataset for subsequent testing. **Open-ended** denotes the capability to provide specific category descriptions for any common or uncommon instance. Open-ended perception is the foundation for subsequent inference and planning in autonomous driving systems. For instance, determining whether an object is collidable, whether it might suddenly move, or whether it signifies that certain areas are not traversable, necessitates an accurate semantic description of the object in the first place.

Many works have been proposed to address these two issues. However, numerous challenges remain when developing open-world perception models. The first challenge in 3D open-world perception for autonomous driving lies in the scarcity of evaluation benchmarks. Specifically, a unified benchmark for domain transfer evaluation is currently absent, and due to the varying formats of individual

Table 1: **Open-world autonomous driving datasets or benchmarks.** "*" means rough estimates. OpenAD is the first real-world open-world benchmark for autonomous driving 3D perception. Compared to other real-world datasets, OpenAD boasts greater category diversity and more instances.

| Datasets | Sensors | Real | Temporal | Scenes | Classes | Instances | GroundTruth |
|---|---|---|---|---|---|---|---|
| GTACrash [31] | Cam. | ✗ | ✔ | 7,720 | 1 | 24K* | **Bbox**(2D) |
| StreetHazards [25] | Cam. | ✗ | ✔ | 1,500 | 1 | 1.5K* | Sem. mask(2D) |
| Synthetic Fire Hydrants [7] | Cam. | ✗ | ✗ | 30,000 | 1 | 30K* | **Bbox**(2D) |
| Synthetic Crosswalks [7] | Cam. | ✗ | ✗ | 20,000 | 1 | 20K* | **Bbox**(2D) |
| CARLA-WildLife [45] | Cam. Depth | ✗ | ✔ | 26 | 18 | 65 | **Inst. mask**(2D) |
| MUAD [19] | Cam. Depth | ✗ | ✗ | 4,641 | 9 | 30K | Sem. mask(2D) |
| AnoVox [5] | Cam. Lidar | ✗ | ✔ | 1,368 | 35 | 1.4K | **Inst.mask**(2D,**3D**) |
| YouTubeCrash [31] | Cam. | ✔ | ✔ | 2,400 | 1 | 12K* | **Bbox**(2D) |
| RoadAnomaly21[12] | Cam. | ✔ | ✗ | 110 | 1 | 0.1K* | Sem. mask(2D) |
| Street Obstacle Sequences [45] | Cam. Depth | ✔ | ✔ | 20 | 13 | 30* | **Inst. mask**(2D) |
| Vistas-NP[21] | Cam. | ✔ | ✗ | 11,167 | 4 | 11K* | Sem. mask(2D) |
| Lost and Found[49] | Cam. | ✔ | ✔ | 112 | 42 | 0.2K* | Sem. mask(2D) |
| Fishyscapes[4] | Cam. | ✔ | ✗ | 375 | 1 | 0.5K* | Sem. mask(2D) |
| RoadObstacle21[12] | Cam. | ✔ | ✔ | 412 | 1 | 1.5K* | Sem. mask(2D) |
| BDD-Anomaly[25] | Cam. | ✔ | ✗ | 810 | 3 | 4.5K | Sem. mask(2D) |
| CODA[33] | Cam. Lidar | ✔ | ✔ | 1,500 | 34 | 5.9K | **Bbox**(2D) |
| OpenAD (ours) | Cam. Lidar | ✔ | ✔ | 2,000 | 206 | 19.8K | **Bbox**(2D,**3D**) |

datasets, researchers must expend considerable effort on the engineering aspect of format alignment. In addition, current 3D perception datasets possess a limited number of common semantic categories, lacking an effective evaluation for open-ended 3D perception models.

The second challenge is the difficulty of training open-world perception models due to the limited scales of publicly available 3D perception datasets. Some open-world language models and 2D perception models have recently used large-scale Internet data for training. How to transfer these models' capabilities to 3D open-world perception is an important and timely research problem.

The last challenge is the relatively low precision of the existing open-world perception models. Specialized models, which lack the capability to understand uncommon objects, exhibit stronger performance for common categories. Consequently, current open-world perception models cannot yet replace specialized models in practice.

To address the aforementioned challenges, we propose OpenAD, an Open-World Autonomous Driving Benchmark for 3D Object Detection. We align the format of five existing autonomous driving perception datasets, select 2,000 scenes, annotate thousands of corner case objects with MLLMs, and develop open-world evaluation metrics to overcome the scarcity of evaluation benchmarks. Then, we introduce a novel vision-centric framework for 3D open-world perception, which utilizes existing 2D open-world perception models to resolve the second challenge. Compared to existing methods, this approach achieves higher average Precision and Recall on the OpenAD benchmark. Finally, we further design a fusion method to address the last challenge by leveraging the strengths of open-world perception models and specialized models to improve the 3D open-world perception results.

The main contributions of this work are:

- We propose an open-world benchmark that simultaneously evaluates object detectors' domain generalization and open-ended capabilities. To our knowledge, this is the first real-world autonomous driving benchmark for 3D open-world object detection.
- We design a labeling pipeline integrated with MLLM, which is utilized to automatically identify corner case scenarios and provide semantic annotations for abnormal objects.
- We propose a novel vision-centric framework for 3D open-world perception. In addition, we analyze the strengths and weaknesses of open-world and specialized models and further introduce a fusion approach to utilize both advantages.

## 2 Related Work

### 2.1 Benchmark for Open-world Object Detection

**2D Benchmark.** Various datasets [38, 23, 52, 34, 20] have been used for 2D open-vocabulary and open-ended object detection evaluation. The most widely used is the LVIS dataset [23], which contains 1,203 categories. In the autonomous driving area, as shown in Table 1, many datasets [25,

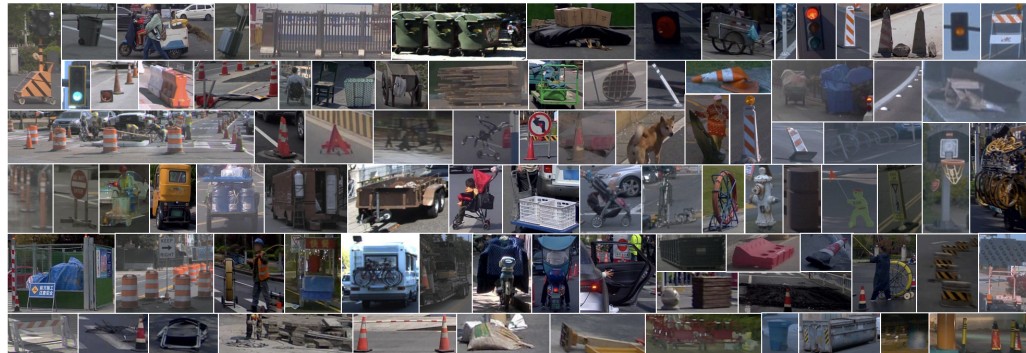

Figure 1: **Examples of corner case objects in OpenAD.** These object categories have not been encountered by models trained on common 3D perception datasets during their training phase.

7, 45, 19, 12, 21, 49, 4, 25, 33] have been proposed. However, some datasets only provide semantic segmentation annotations without specific instances or annotate objects as abnormal but lack semantic tags. Moreover, datasets collected from real-world driving data are on a small scale, while synthetic data from simulation platforms such as CARLA [17] lack realism, making it difficult to conduct effective evaluations. In contrast, our OpenAD offers large-scale 2D and 3D bounding box annotations from real-world data for a more comprehensive open-world object detection evaluation.

**3D Benchmark.** The 3D open-world benchmarks can be divided into two categories: indoor and outdoor scenarios. For indoor scenarios, SUN-RGBD [53] and ScanNet [16] are two real-world datasets often used for open-world evaluation, containing about 700 and 21 categories, respectively. For outdoor or autonomous driving scenarios, AnoVox [5] is a synthetic dataset, containing instance masks of 35 categories for open-world evaluation. However, due to limited simulation assets, the quality and instance diversity of the synthetic data are inferior to real-world data. Existing real-data 3D object detection datasets for autonomous driving [8, 46, 57, 54, 20] only contain a few common object categories, which can hardly be used to evaluate open-world models. To address these issues, OpenAD is constructed from real-world data and contains 206 different corner-case object categories that appeared in autonomous driving scenarios.

Additionally, the metrics of existing benchmarks are not designed for open-world detection, as their ground-truth annotations presuppose fixed categories. Evaluation inaccuracies may arise when semantic labels overlap or when open-ended models predict synonymous terms. Furthermore, long-tail objects can disproportionately skew the computation of these metrics. OpenAD has redesigned metrics to address this issue.

## 2.2   2D Open-world Object Detection Methods

To address out-of-distribution (OOD) or anomaly detection, earlier approaches [63] typically employed decision boundaries, clustering, etc., to discover OOD objects. Recent methods [30, 71, 44, 41, 59, 61, 68, 34, 55, 70, 14, 58, 22] employ text encoders, such as CLIP [51], to align the text features of the corresponding category labels with the box features. Specifically, GLIP [34] unifies object detection and phrase grounding for pre-training. OWL-ViT v2 [47] uses a pretrained detector to generate pseudo labels on image-text pairs to scale up detection data for self-training. YOLO-World [14] adopts a open-vocabulary YOLO-type architecture and achieves good efficiency. However, all these methods require predefined object categories during inference.

More recently, some open-ended methods [15, 66, 39] utilize natural language decoders to provide language descriptions and facilitate generating category labels from RoI features directly. More specifically, GenerateU [15] introduces a language model to generate class labels directly from regions of interest. DetClipv3 [66] introduced an object captioner to generate class labels during inference and image-level descriptions for training. VL-SAM [39] introduces a training-free framework with the attention map as prompts.

## 2.3   3D Open-world Object Detection Methods

In contrast to 2D open-world object detection tasks, 3D open-world object detection tasks are more challenging due to the limited training datasets and complex 3D environments. To alleviate this issue,

most existing 3D open-world models leverage power of pretrained 2D open-world models or utilize abundant 2D training datasets.

For instance, some indoor 3D open-world detection methods like OV-3DET [43], INHA [29], and ImOV3D [65] use a pretrained 2D object detector to guide the 3D detector to find novel objects. Similarly, Coda [9] utilizes 3D box geometry priors and 2D semantic open-vocabulary priors to generate pseudo 3D box labels of novel categories. FM-OV3D [69] utilizes stable diffusion to generate data containing OOD objects. As for outdoor methods, FnP [18] uses region VLMs and a Greedy Box Seeker to generate annotations for novel classes during training. OV-Uni3DETR [56] utilizes images from other 2D datasets and 2D bounding boxes or instance masks generated by an open-vocabulary detector.

However, these existing 3D open-vocabulary detection models require predefined object categories during inference. To address this issue, we introduce a vision-centric open-ended 3D object detection method, which can directly generate effectively unlimited category labels during inference.

## 3 Properties of OpenAD

### 3.1 Scenes and Annotation

The 2,000 scenes in OpenAD are carefully selected from five large-scale autonomous driving perception datasets: Argoverse 2 [57], KITTI [20], nuScenes [8], ONCE [46] and Waymo [54], as illustrated in Figure 2. These scenes are collected from different countries and regions, and have different sensor configurations. Each scene has temporal camera and LiDAR inputs and contains at least one corner case object that the original dataset has not annotated.

For 3D bounding box labels, we annotate 6,597 corner case objects across these 2,000 scenarios, combined with the annotations of 13,164 common objects in the original dataset, resulting in 19,761 objects in total. The location and size of all objects are manually annotated using 3D and 2D bounding boxes, while their semantic categories are labeled with natural language tags, which can be divided into 206 classes. We illustrate some corner case objects in Figure 1. OpenAD encompasses both abnormal forms of common objects, such as bicycles hanging from the rear of cars, cars with doors open, and motorcycles with rain covers, as well as uncommon objects, including open manhole covers, cement blocks, and tangled wires scattered on the ground.

In addition, we have annotated each object with a "seen/unseen" label, indicating whether the categories of the objects have appeared in the training set of each dataset. This label is intended to facilitate the evaluation process by enabling a straightforward separation of objects that the model has encountered (seen) and those it has not (unseen), once the training dataset is specified. In addition, we offer a toolkit code that consolidates scenes from five original datasets into a unified format, converts them to OpenAD data, and facilitates the loading and visualization process.

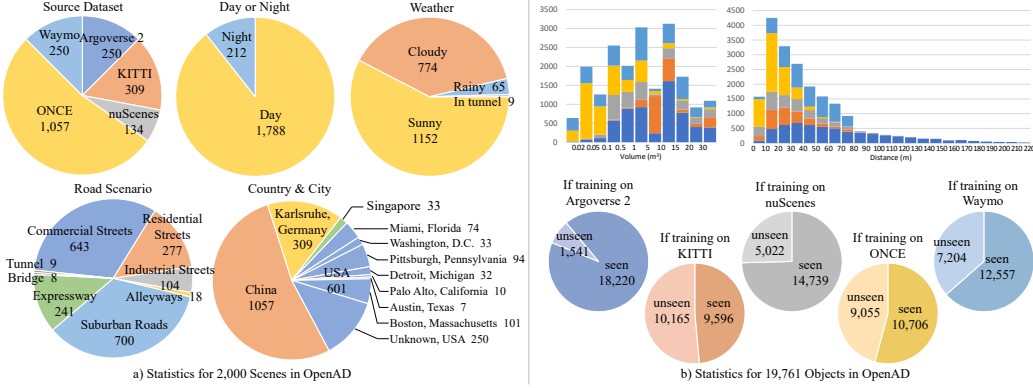

Figure 2: **Data composition of OpenAD.** OpenAD covers multiple cities in various countries, including scenes during the day and night, on different weather and road scenarios. Additionally, we annotate each object with an indication of whether its category is observed in the training set of each dataset, allowing for separate evaluations of the model's specialized performance and open-ended performance.

## 3.2 Evaluation Metrics

In contrast to existing benchmarks, we utilize natural language annotations and conduct multi-threshold matching against string-formatted model predictions. This evaluation framework not only resolves synonym challenges but also demonstrates stronger consistency with the conceptual framework of Open-World paradigms.

**Average Precision (AP) and Average Recall (AR).** The calculation of AP and AR depends on True Positive (TP). In OpenAD, the threshold of TP incorporates both positional and semantic scores. An object prediction is considered a TP only if it simultaneously meets both the positional and semantic thresholds. For 2D object detection, in line with COCO, Intersection over Union (IoU) is used as the positional score. We encode the ground truth text and prediction text using the CLIP model's text encoder and compute the cosine similarity of their text features as the semantic score. When calculating AP, IoU thresholds ranging from 0.5 to 0.95 with a step size of 0.05 are used, along with semantic similarity thresholds of 0.5, 0.7, and 0.9.

For 3D object detection, the center distance is adopted as the positional score following nuScenes, and we use the same semantic score as the 2D detection task. Similar to nuScenes, we adopt a multi-threshold averaging method for AP calculation. Specifically, it computes AP across 12 thresholds, combining positional thresholds of 0.5m, 1m, 2m, and 4m with semantic similarity thresholds of 0.5, 0.7, and 0.9, and then average these AP values.

The same principle applies to calculating Average Recall (AR) for 2D and 3D object detection tasks. Both AP and AR are calculated only for the top 300 predictions.

**Average Translation Error (ATE) and Average Scale Error (ASE).** Following nuScenes, we also evaluate the prediction quality of TP objects using regression metrics. The Average Translation Error (ATE) refers to the Euclidean center distance, measured in pixels for 2D or meters for 3D. The Average Scale Error (ASE) is calculated as $1 - IoU$ after aligning the centers and orientations of the predicted and ground truth objects.

**In/Out Domain & Seen/Unseen AR.** To evaluate the model's domain generalization ability and open-ended capability separately, we calculate the AR based on whether the scene is within the training domain and whether the object semantics have been seen during training. The positional thresholds for this metric are defined as above, whereas the semantic similarity thresholds are fixed at 0.9.

## 4 Construction of OpenAD

We propose a vision-centric semi-automated annotation pipeline for OpenAD, as shown in Figure 3. This differs from the existing LiDAR-based pipeline [33] because certain objects, such as cables or nails close to the road surface and wall-hung signboards, cannot be detected solely by LiDAR.

We use an MLLM Abnormal Filter to identify scenes containing corner cases within the validation and test sets of five autonomous driving datasets, followed by manual filtering. We then annotate the corner case objects with 2D bounding boxes.

For objects with relatively complete 3D geometry formed by point clouds, we adopt point-cloud-clustering algorithms [6] to generate candidate 3D bounding boxes. We utilize camera parameters to project 2D bounding boxes into the point cloud space and identify the corresponding clusters.

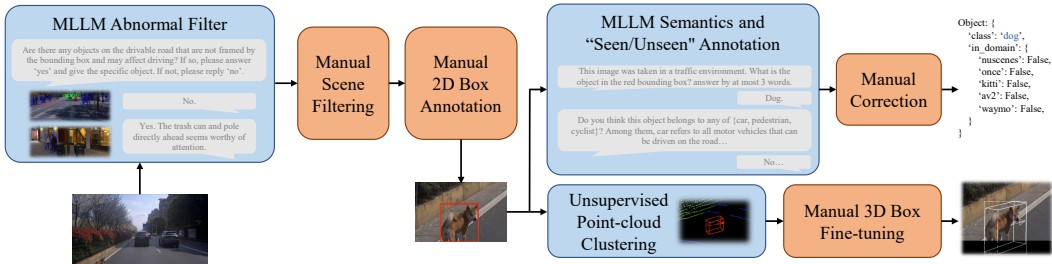

Figure 3: **Annotation pipeline**. OpenAD is built upon a corner case discovery and annotation pipeline that integrates with a multimodal large language model (MLLM).

Finally, the bounding boxes are manually corrected. For objects without corresponding 3D candidates matching their 2D bounding boxes, we manually annotate 3D bounding boxes by referencing multi-view images.

For category labels, we feed images with 2D bounding boxes to an MLLM for semantic annotation and indicate for each object whether its category has been seen in each dataset. To select the best MLLM and prompts for object recognition, we manually select 30 challenging annotated image samples and evaluate the accuracy of each MLLM and prompt. We tried GPT-4V [48], Claude 3 Opus [2], and InternVL 1.5 [13], with InternVL exhibiting the best performance. Our experiments also reveal that closed image prompts, such as 2D bounding boxes or circles, yield the best results, whereas marking the object of inquiry on the image with arrows yields slightly inferior results. Objects such as open manholes and wires falling on the road are difficult to identify for existing MLLMs. This implies that OpenAD can also be utilized to test the detection and semantic recognition capabilities of MLLMs. Appendix A presents a detailed demonstration of how we improved our pipeline. The final MLLM and prompt achieve an accuracy of approximately 90% on the entire dataset.

Note that although we have utilized tools such as MLLM to automate some stages as much as possible to reduce manual workload, we have also incorporated manual verification into each stage to ensure the accuracy of each annotation.

## 5  Baseline Methods of OpenAD

### 5.1  Vision-Centric 3D Open-ended Object Detection

Due to the limited scale of existing 3D perception data, it is challenging to train a vision-based 3D open-world perception model directly. To address this issue, we propose a vision-centric framework for 3D open-world perception, as illustrated in Figure 4. An existing 2D open-world object detection model is first employed to generate 2D proposals and their corresponding semantic labels. Subsequently, a 2D-to-3D BBox Converter is introduced, which combines multiple features and a few trainable parameters, to transform 2D proposals into 3D boxes.

Specifically, for each 2D proposal, we employ a Partial Encoder composed of multi-layer convolutional networks to extract partial features. This is followed by a Depth Net constructed with MLPs to predict depth at each grid, ultimately generating a depth map. We also include an optional branch that utilizes LiDAR point clouds and a linear fitting function to refine the depth map by projecting point clouds onto the image. A 3D point coordinate can be obtained from the coordinates of each grid, camera parameters, and depth. Pseudo point clouds with features are obtained in this way. We project the pseudo point clouds onto the feature map and depth map, and features are assigned to each point through interpolation. Then, we adopt PointNet [50] to extract the feature $f_p$ of the pseudo point clouds. Meanwhile, the depth and feature maps within the 2D bounding box are concatenated along the channel dimension, and its feature $f_c$ is derived through convolution and global pooling. Finally, we utilize an MLP to predict the object's 3D bounding box with the concatenated features of $f_p$ and $f_c$.

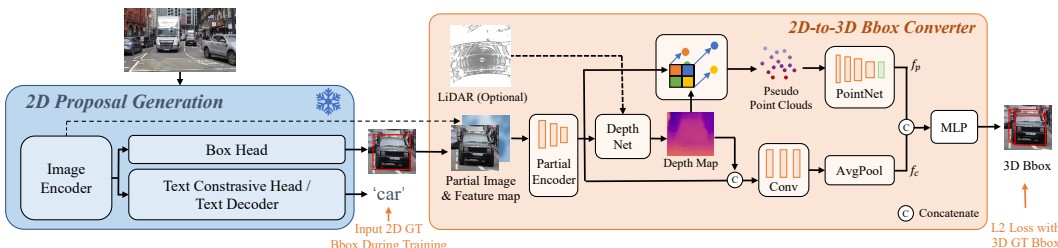

Figure 4: **The 3D open-world object detection framework we proposed**. After obtaining 2D proposals from any frozen open-world 2D object detection model, we train a 2D-to-3D BBox Converter to predict 3D bounding boxes. The converter has a dual-branch architecture, which extracts pseudo-point features and convolutional features. It is lightweight and easy to train.

Only a few parameters in the 2D-to-3D Bbox Converter are trainable in this baseline. Thus, the training cost is low. In addition, during the training, each 3D object serves as a data point for this baseline, allowing for the straightforward construction of multi-domain dataset training.

## 5.2 General and Specialized Models Fusion

In experiments, we find that existing open-world methods or general models are inferior to close-set methods or specialized models in handling objects belonging to common categories, but they exhibit stronger domain generalization capabilities and the ability to deal with corner cases. That is to say, existing general and specialized models complement each other. Hence, we utilize their strengths and propose a fusion baseline by combining the prediction results from the two types of models. Specifically, we align the confidence scores of the two types of models and perform non-maximum suppression (NMS) with dual thresholds, *i.e.*, IoU and semantic similarity, to filter duplicates.

# 6 Experiments

## 6.1 Implementation Details

For specialized models that can only predict common categories, we directly match their prediction results with the corresponding categories and sort them according to their confidence scores.

Open-ended 2D methods can directly provide bounding boxes and the corresponding natural language descriptions, enabling direct evaluation of OpenAD. For 2D open-vocabulary methods, which need a predefined object category list from users as additional inputs to detect corresponding objects, we take the union of the common categories from five datasets and incorporate two additional open-vocabulary queries, *i.e.*, "object that affects traffic" and "others", into it. For all open-world 2D models, we selected their best-performing open-source model and conducted zero-shot evaluation on OpenAD.

For 3D Open-vocabulary methods, the original version of Find n' Propagate [18] utilizes a 2D detector trained on the full nuScenes dataset to provide pseudo-labels. For a fair comparison, we employ YOLO-world v2 to provide the pseudo-labels instead.

For the baseline method we proposed, the 2D-to-3D Converter is trained on nuScenes (10 common classes only) for 10 epochs, over 20 hours using 4 A40 GPUs. We use GenerateU [15] and YOLO-World [14] to generate the 2D proposals, respectively. They are frozen without any fine-tuning. We employ Swin-tiny [42] as the Partial Encoder and a multi-layer convolutional network as the Depth Net. The partial images have a resolution of 224*224.

## 6.2 Main Results

Tables 2 and 3 show the evaluation results on 2D and 3D object detection models, including 2D and 3D open-world models, specialized models, and our baselines. The results show that current open-world models, irrespective of whether they are 2D or 3D detectors, tend to predict objects

Table 2: **Evaluation of 2D open-world methods (top), specialized methods (middle), and ensemble methods (bottom) on OpenAD benchmark.** $AR^{nusc}$ refers to scenes derived from nuScenes in OpenAD, with $AR_{seen}$ denoting object categories observed in the nuScenes training set. For 2D open-world methods, we utilize open-source models for zero-shot inference, but for comparison purposes, classification AR against nuScenes is also presented. All specialized methods are trained on nuScenes.

| Method | Backbone/Base-model | AP↑ | AR↑ | ATE↓ | ASE↓ | $AR^{nusc}_{seen}$ ↑ | $AR^{nusc}_{unseen}$ ↑ | $AR^{others}_{seen}$ ↑ | $AR^{others}_{unseen}$ ↑ |
|---|---|---|---|---|---|---|---|---|---|
| GLIP [34] | Swin-L | 7.14 | 16.01 | 6.581 | 0.1352 | 1.83 | 1.28 | 2.33 | 1.05 |
| VL-SAM [39] | ViT-H | 8.46 | 17.50 | 6.630 | 0.1355 | 9.66 | 5.41 | 9.13 | 3.43 |
| OWL-ViT v2 [47] | ViT-L | 9.70 | 21.17 | 6.284 | 0.1461 | 21.42 | 4.66 | 18.97 | **8.01** |
| GenerateU [15] | Swin-L | 9.77 | 21.75 | 6.743 | 0.1360 | 12.74 | 7.18 | 18.79 | 7.31 |
| YOLO-World v2 [14] | YOLOv8-X | 10.20 | 23.46 | 7.489 | 0.1397 | 18.68 | **10.16** | 20.61 | 7.27 |
| GroundingDino [41] | Swin-L | 8.52 | 26.67 | 6.499 | 0.1432 | 20.53 | 4.21 | 21.26 | 7.36 |
| MaskRCNN [24] | ResNet50 | 12.76 | 20.07 | 6.126 | 0.1359 | 27.77 | 0.00 | 23.41 | 0.07 |
| MaskRCNN [24] | VovNetv2-99 | 12.32 | 21.09 | 5.746 | 0.1338 | 30.21 | 0.00 | 21.74 | 0.09 |
| DETR [10] | ResNet50 | 12.46 | 20.35 | 6.066 | 0.1346 | 28.27 | 0.00 | 21.35 | 0.03 |
| DINO [11] | ResNet50 | 15.24 | 23.41 | 5.679 | **0.1258** | 35.49 | 0.00 | 26.39 | 0.02 |
| Co-DETR [72] | ResNet50 | 15.65 | 24.63 | 5.421 | 0.1270 | 38.82 | 0.00 | 27.96 | 0.03 |
| Co-DETR [72] | Swin-L | 16.21 | 27.76 | **5.386** | 0.1287 | 45.41 | 0.00 | 26.14 | 0.01 |
| OpenAD-Ens | YOLO-world + MaskRCNN(V2-99) | 13.28 | 29.74 | 6.726 | 0.1409 | 33.30 | 10.05 | 26.92 | 7.20 |
| OpenAD-Ens | YOLO-world + Co-DETR(Swin-L) | **16.94** | **34.38** | 6.457 | 0.1368 | **46.65** | 10.06 | **30.39** | 7.20 |

Table 3: **Evaluation of 3D open-world methods (top), specialized methods (middle), and ensemble methods (bottom) on OpenAD benchmark.** $AR^{nusc}$ refers to scenes derived from nuScenes in OpenAD, with $AR_{seen}$ denoting object categories observed in the nuScenes training set. All methods are trained on nuScenes training set.

| Method | Modality | Backbone/Base-model | AP↑ | AR↑ | ATE↓ | ASE↓ | $AR^{nusc}_{seen}$ ↑ | $AR^{nusc}_{unseen}$ ↑ | $AR^{others}_{seen}$ ↑ | $AR^{others}_{unseen}$ ↑ |
|---|---|---|---|---|---|---|---|---|---|---|
| OpenAD-G | C | GenerateU | 6.01 | 12.90 | 1.342 | 0.504 | 11.35 | 3.64 | 15.18 | 3.71 |
| OpenAD-Y | C | YOLOWorld | 6.26 | 13.89 | 1.338 | 0.487 | 14.64 | 7.18 | 18.79 | 3.53 |
| FnP [18] | L | SECOND | 8.85 | 18.97 | 0.848 | 0.493 | 18.49 | 10.82 | 23.42 | 7.47 |
| OpenAD-G | LC | GenerateU | 15.14 | 34.46 | 1.056 | 0.649 | 14.54 | 11.15 | 26.48 | **16.95** |
| OpenAD-Y | LC | YOLOWorld | 15.54 | 36.07 | 1.063 | 0.646 | 29.99 | **12.73** | 25.88 | 14.17 |
| BEVDet [27] | C | ResNet50 | 9.42 | 15.63 | 1.183 | 0.438 | 36.46 | 0.00 | 14.11 | 0.00 |
| BEVFormer [36] | C | ResNet50 | 10.08 | 19.36 | 1.125 | 0.440 | 39.38 | 0.00 | 15.85 | 0.00 |
| BEVFormer [36] | C | ResNet101-DCN | 14.43 | 22.73 | 0.978 | 0.444 | 51.86 | 0.00 | 16.59 | 0.03 |
| BEVDepth4D [26] | C | ResNet50 | 12.33 | 20.70 | 1.118 | 0.480 | 39.75 | 0.00 | 17.94 | 0.02 |
| BEVStereo [35] | C | ResNet50 | 11.12 | 18.27 | 1.133 | 0.431 | 36.73 | 0.00 | 16.21 | 0.00 |
| BEVStereo [35] | C | VovNetv2-99 | 10.58 | 16.03 | 1.118 | 0.388 | 51.69 | 0.00 | 13.05 | 0.01 |
| HENet [60] | C | Vov2-99 + R50 | 11.58 | 17.48 | 1.070 | **0.386** | 52.02 | 0.00 | 14.65 | 0.01 |
| SparseBEV [40] | C | ResNet50 | 7.61 | 16.97 | 1.142 | 0.435 | 60.04 | 0.00 | 7.48 | 0.02 |
| SparseBEV [40] | C | VovNetv2-99 | 7.64 | 16.93 | 1.103 | 0.431 | 61.36 | 0.00 | 7.09 | 0.01 |
| BEVFormer v2 [62] | C | ResNet50 | 14.64 | 33.13 | 1.064 | 0.554 | 56.63 | 0.00 | 27.16 | 0.08 |
| Centerpoint [67] | L | SECOND | 13.79 | 26.79 | 0.667 | 0.499 | 44.23 | 0.00 | 11.42 | 0.04 |
| TransFusion-L [3] | L | SECOND | 14.64 | 34.02 | **0.653** | 0.655 | 52.18 | 0.00 | 24.02 | 0.00 |
| BEVFusion [37] | LC | SECOND + Dual-Swin-T | 15.57 | 33.50 | 0.730 | 0.449 | 59.93 | 0.00 | 20.64 | 0.00 |
| OpenAD-Ens | C | OpenAD-Y + HENet | 12.36 | 24.32 | 1.176 | 0.420 | 54.16 | 7.18 | 23.37 | 3.53 |
| OpenAD-Ens | LC | FnP + BEVFusion | 16.19 | 42.08 | 0.776 | 0.458 | 61.74 | 10.82 | 28.40 | 7.47 |
| OpenAD-Ens | LC | OpenAD-Y + BEVFusion | 16.22 | 47.12 | 0.851 | 0.511 | 62.69 | 12.05 | 35.62 | 13.60 |
| OpenAD-Ens | LC | OpenAD-G + BEVFusion | **16.30** | **48.25** | 0.858 | 0.520 | **64.84** | 10.59 | **39.11** | 16.85 |

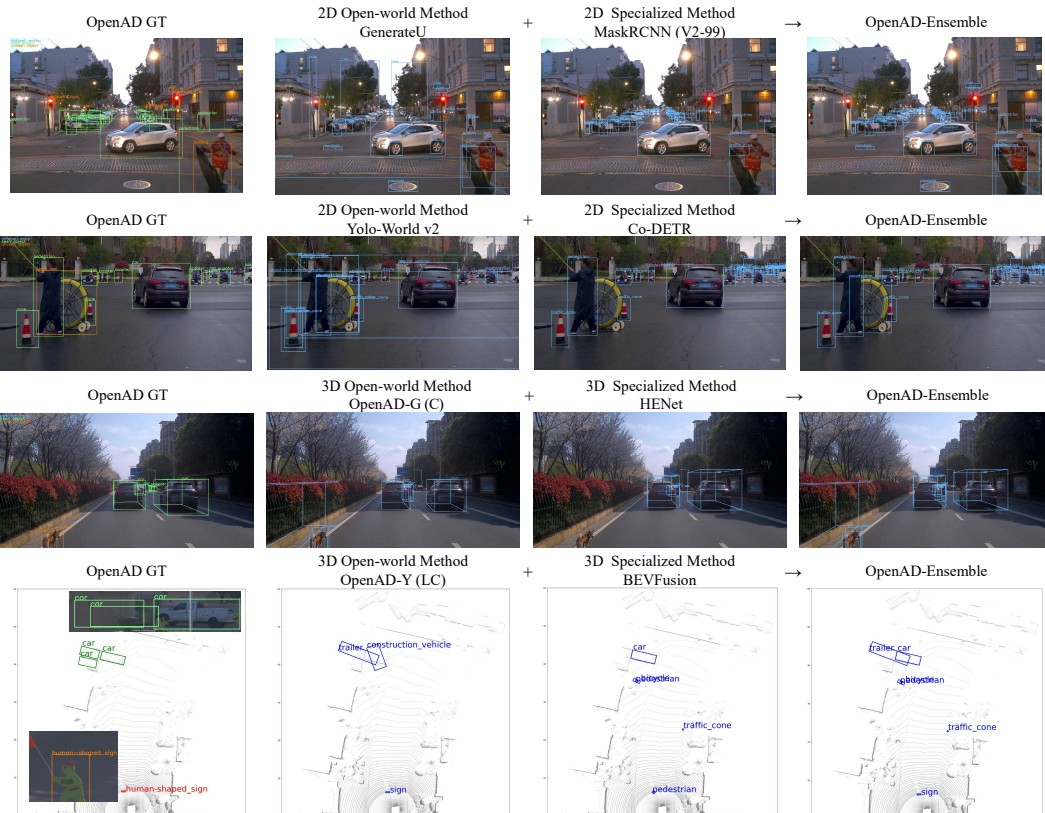

Figure 5: **Example results of open-world models, specialized models, and our proposed ensemble method**.

unrelated to driving (such as the sky) or to make repeated predictions for different parts of the same object, resulting in low precision and AP. However, these models demonstrate good domain generalization and open-vocabulary capabilities, which are lacking in current specialized models. Note that our proposed ensemble baselines can effectively combine the advantages of open-world and specialized models, achieving favorable performance in both seen and unseen domains and categories. In addition, in Table 3, our proposed vision-centric baseline for 3D open-world object detection utilizes the capabilities of 2D open-world models. Specifically, by harnessing the open-

Table 4: **Ablation of 2D-to-3D BBox Converter**. This module is trained on nuScenes training set and tested on OpenAD. The 2D proposals are generated by GenerateU [15].

| Conv & AvgPool | Pseudo Point Clouds | Depth | Bbox Decoding | AP | AR |
|:---:|:---:|:---:|:---:|:---:|:---:|
| ✔ | ✗ | Frozen Depth Anything | MLP | 7.90 | 18.38 |
| ✗ | ✔ | Frozen Depth Anything | PCA for Oriented Bounding Box | 7.61 | 13.63 |
| ✗ | ✔ | Frozen Depth Anything | MLP | 8.97 | 22.02 |
| ✔ | ✔ | Frozen Depth Anything | MLP | 9.02 | 23.32 |
| ✔ | ✔ | Trainable Depth Net | MLP | 15.14 | 34.46 |

world capabilities of Yolo-world v2, our method obtains 6.7 AP and 17.1 AR improvement compared to Find n' Propagate.

Moreover, we observed that the issue of overfitting is more pronounced for 3D object detection models on datasets such as nuScenes. Some models perform well in-domain benchmarks but show worse domain generalization ability. For instance, SparseBEV, compared to methods based on Lift-Splat-Shot, achieves impressive in-domain results, with its in-domain AR even surpassing those of LiDAR-based methods. However, SparseBEV's domain generalization capability is relatively poor. Models with increased parameters by enlarging the backbone, including BEVStereo and SparseBEV, show more severe overfitting issues. These results reveal the limitations of in-domain benchmarks such as nuScenes. In contrast, increasing parameters through utilizing BEVFormer v2 or HENet simultaneously enhances both in-domain and out-domain Recall, indicating an inherent improvement in the methodology. Therefore, even for specialized models trained on a single domain, evaluating them on OpenAD benchmarks remains meaningful.

Furthermore, as shown in Figure 5, we provide visualization samples for some methods. Objects enclosed by orange bounding boxes belong to unseen categories in nuScenes. Recognition of these objects relies on open-world models. In contrast, specialized models exhibit significant advantages for common objects, especially for distant objects.

The proposed converter enables convenient cross-dataset training. As detailed in Appendix B, AP and AR of our proposed method can be further improved by training on multiple datasets and combining 2D segmentation models.

### 6.3 Ablations of Proposed Method

We conduct ablation studies for the proposed baselines, as shown in Table 4. When decoding bounding boxes solely from either the pseudo point features $f_p$ or the convolutional features $f_c$, performance drops, demonstrating the effectiveness of our proposed dual-branch architecture. In addition, replacing MLP with unlearnable PCA methods decreases the performance by a large margin, from 23.32 AR to 13.63 mAR. These results show that the simple MLP can learn to complete the boundaries of objects from the datasets and predict more accurate 3D boxes. In initial experiments, we employed a frozen Depth Anything [64] model to obtain depth estimations. Subsequent experimental results reveal that using a lightweight trainable depth network can enhance the converter's performance.

Section 5.2 focuses on General and Specialized Fusion, not mere ensemble methods. Ensemble techniques serve only as a preliminary means to implement General-Specialized Fusion. For comparison, we present in Table 5 results from: ensembles of two general models, and ensembles of two specialized models. Demonstrably, General-Specialized Fusion exhibits marked superiority over these normal ensembles. Most evidently, while normal ensembles employ NMS for deduplication, they struggle to achieve consistent AP improvement. In contrast, General-Specialized Fusion effortlessly enhances AP while delivering substantially greater gains in AR.

## 7 Limitations

OpenAD exclusively supports 2D and 3D object detection tasks, with all re-annotated data allocated for benchmarking purposes. In the future, we will incorporate evaluations for additional open-world perception tasks, such as occupancy prediction, while expanding data to enhance scope and diversity.

Table 5: **Ablation of General-Specialized Fusion**. General-Specialized Fusion exhibits marked superiority over normal ensembles.

| No. | Method | AP↑ | AR↑ | $AR_{seen}^{nusc}$ ↑ | $AR_{unseen}^{nusc}$ ↑ | $AR_{seen}^{others}$ ↑ | $AR_{unseen}^{others}$ ↑ |
|---|---|---|---|---|---|---|---|
| G1 | YoloWorld + Converter | 15.54 | 36.07 | 29.99 | 12.73 | 25.88 | 14.17 |
| G2 | VL-SAM + Converter | 18.60 | 39.16 | 16.63 | 15.77 | 29.28 | 20.60 |
| G1+G2 | NMS | 15.75↓ | 43.94↑ | 33.81 | 21.51 | 34.86 | 24.61 |
| S1 | BEVFormer | 14.43 | 22.73 | 51.86 | 0.00 | 16.59 | 0.03 |
| S2 | BEVFusion | 15.57 | 33.50 | 59.93 | 0.00 | 20.64 | 0.00 |
| S1+S2 | NMS | 14.54↓ | 38.37↑ | 64.29 | 0.00 | 27.37 | 0.03 |
| G1+S2 | OpenAD-Ens | 16.30↑ | 48.25↑ | 64.84 | 10.59 | 39.11 | 16.85 |
| G2+S1 | OpenAD-Ens | 18.74↑ | 44.65↑ | 57.54 | 15.77 | 35.68 | 20.63 |
| G2+S2 | OpenAD-Ens | 18.90↑ | 50.99↑ | 65.10 | 15.77 | 41.86 | 20.60 |

## 8   Conclusion

In this paper, we introduce OpenAD, the first open-world autonomous driving benchmark for 3D object detection. OpenAD is built upon a corner case discovery and annotation pipeline that is integrated with a multimodal large language model. The pipeline aligns five autonomous driving perception datasets in format and annotates corner case objects for 2000 scenarios. In addition, we devise evaluation methodologies and analyze the strengths and weaknesses of existing open-world perception models and specialized autonomous driving perception models. Moreover, to address the challenge of training 3D open-world models, we propose a novel framework for 3D open-world perception, which is lightweight and easy to train. Furthermore, we introduce a fusion baseline approach to take advantage of open-world and specialized models.

Through evaluations conducted on OpenAD, we have observed that existing open-world models are still inferior to specialized models within the in-domain context, yet they exhibit stronger domain generalization and open-vocabulary abilities. It is worth noting that the improvement of certain models on in-domain benchmarks comes at the expense of their open-world capabilities, while this is not the case for other models. This distinction cannot be revealed solely by testing on in-domain benchmarks.

## Acknowledgements

This work was supported by National Key R&D Program of China (Grant No. 2022ZD0160305).

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

# A. Ablation on the Annotation Pipeline.

As shown in Figure 6, we conduct experiments by employing diverse visual and textual prompts, along with various MLLMs, and select the optimal approach.

The experimental results concerning visual prompts indicate that, as object cues, squares outperform ellipses, while arrows perform less satisfactorily than both. Notably, it is crucial for objects located at the edges of images to maintain the closure of their bounding squares.

Objects like open manholes and wires falling on the road are difficult to identify for existing MLLMs. For such objects, MLLMs tend to respond with other nearby objects. Requiring existing MLLMs to rethink may still not improve the accuracy of their responses.

We use GPT-4V [48], Claude 3 Opus [2], and InternVL 1.5 [13], with InternVL exhibiting the best performance. This may be because InternVL has been trained on more autonomous driving data.

Accuracy is manually calculated based on five repetitions of testing on 30 highly challenging samples. During the manual verification of automated annotations, we conducted a preliminary assessment of the accuracy of the pipeline. The final MLLM and prompt achieve an accuracy rate of approximately 90% on the entire OpenAD data.

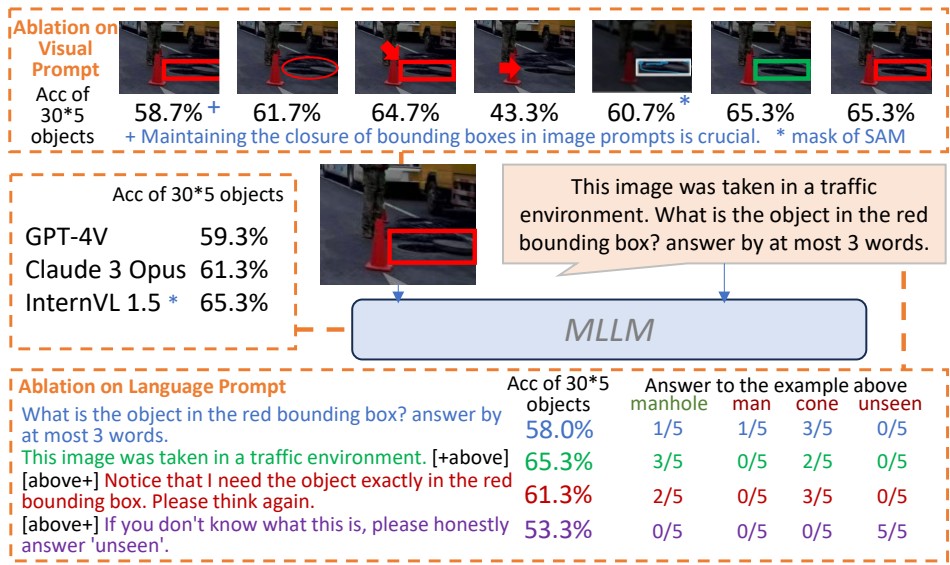

Figure 6: **Ablation on annotation pipeline.** We conduct experiments by employing diverse visual and textual prompts, along with various MLLMs, and select the optimal approach. Accuracy is manually calculated based on five repetitions of testing on 30 highly challenging samples.

# B. Further Enhance the Performance of the Proposed Baseline.

## B.1 Cross-dataset Training.

Since the proposed converter's training relies on 2D-3D ground truth bounding box pairs, our framework enables convenient cross-dataset training. Table 6 shows that training on three datasets (common classes only) leads to performance gains.

## B.2 Using an Instance Segmentation Model.

Some pseudo point clouds generated from background pixels (e.g., road surface within bounding boxes) may introduce noise. To eliminate this noise, we utilize the Segment Anything Model [32]

Table 6: **Performance Comparison: Single Dataset vs. Cross-Dataset Training.** AP and AR of our proposed method can be further improved by training on multiple datasets.

| Method | Training on | AP↑ | AR↑ | ATE↓ | ASE↓ |
|---|---|---|---|---|---|
| OpenAD-G | nuScenes | 15.14 | 34.46 | 1.056 | 0.649 |
| OpenAD-G | nuScenes + Waymo + KITTI | 19.42 | 38.08 | 0.926 | 0.662 |
| OpenAD-Ens | nuScenes | 16.30 | 48.25 | 0.858 | 0.520 |
| OpenAD-Ens | nuScenes + Waymo + KITTI | 19.72 | 53.41 | 0.869 | 0.546 |

Table 7: **Performance Comparison: With vs. Without Segmentation**. This module is trained on nuScenes training set and tested on OpenAD. The 2D proposals are generated by GenerateU [15]. Segmentation results are derived from Segment Anything [32].

| Depth | Segmentation | AP | AR |
|---|---|---|---|
| Frozen Depth Anything | ✗ | 9.02 | 23.32 |
| Frozen Depth Anything | ✔ | 9.07 | 24.09 |

(SAM) to segment the object with the 2D box as the prompt, yielding a segmentation mask. While using SAM can bring about marginal improvements, it bloats the framework. Therefore, we have excluded segmentation from the latest version of our baseline. However, if the 2D model used in the framework inherently supports instance segmentation (e.g., VL-SAM [39]), this performance gain can be achieved without additional computational overhead.

## C. More Statistics on OpenAD Data.

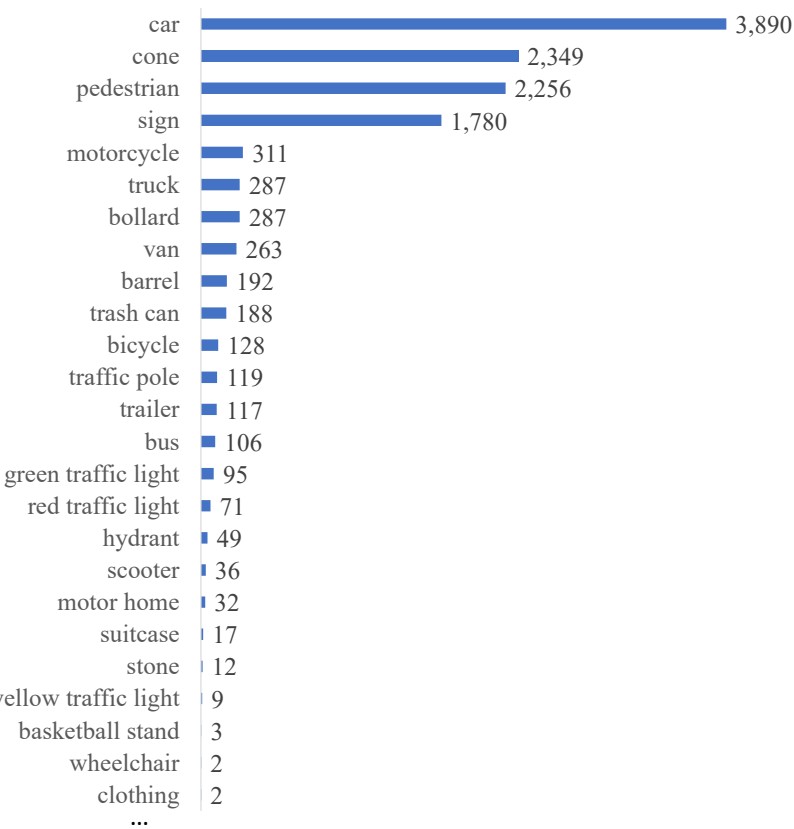

Figure 7: **Statistics on the number of objects in certain categories in OpenAD.**

Since OpenAD is designed to evaluate a model's ability to understand unknown objects, we cannot disclose all category labels in OpenAD. However, we provide quantitative statistics for a subset of labels (common objects or those illustrated in the sample data), as shown in Figure 7. Additionally, Figure 1 demonstrates the diversity of the OpenAD dataset. Figure 8 also shows images of some rare objects and their corresponding LiDAR point clouds in OpenAD.

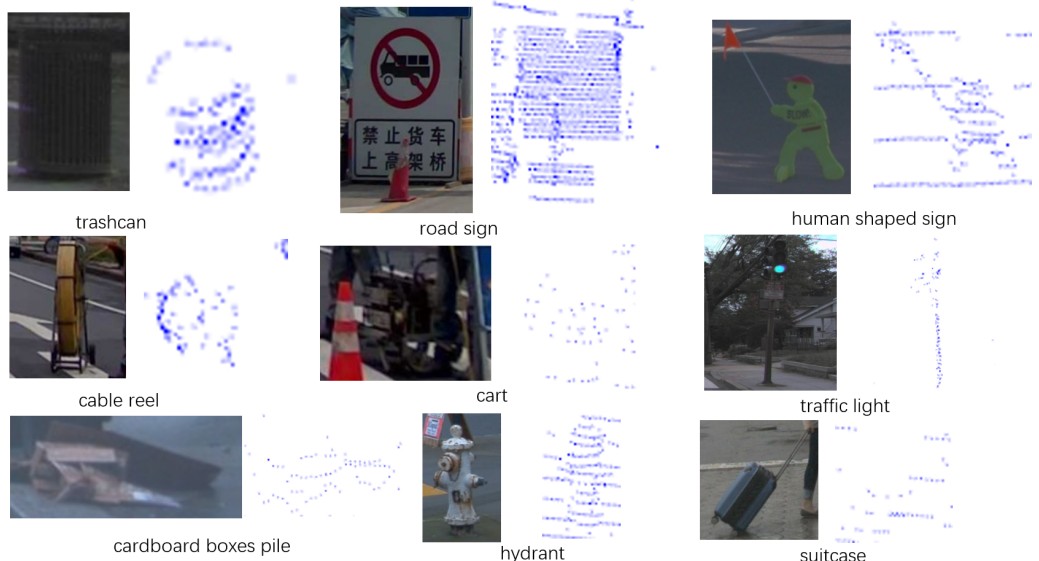

Figure 8: **Examples of corner case objects in OpenAD.** These object categories have not been encountered by models trained on common 3D perception datasets during their training phase.

## D. Broader Impacts Statement.

All data utilized in OpenAD are sourced from published datasets. We do not see potential privacy-related issues. This study may inspire future research on open-world perception models.

