# OpenReview forum: "OpenAD: Open-World Autonomous Driving Benchmark for 3D Object Detection"
_NeurIPS.cc/2025/Datasets_and_Benchmarks_Track — NeurIPS 2025 Datasets and Benchmarks Track poster_

### Official Review · Reviewer_QGow · 2025-06-25

**Rating:** 4
**Confidence:** 4

**Summary:**

This paper proposes an open-world autonomous driving benchmark for 3D object detection. Built on corner case discovery, the annotation pipeline aligns five AD perception datasets in format and annotates corner case objects for 2000 scenarios. Evaluation methodologies are devised to evaluate the existing perception models. Moreover, this paper also proposes a novel framework for 3D open-world perception and a fusion baseline approach to take advantage of open-world and specialized models.

**Dataset Code Accessibility:**

Yes

**Ethical Considerations:**

No, there are no or only very minor ethics concerns

**Final Justification:**

My concerns have been addressed mostly. I choose to keep my positive score.

**Limitations Weaknesses:**

1.	The proposed object detection framework has been investigated form previous 3D object detection approaches.
2.	For the tasks of domain generalization and open-ended detection, the contribution of different methods to this issue will vary. Specify dataset construction and experiments are required to see what really matters for domain generalization or open-ended detection.
3.	The ensemble of baselines naturally brings performance gains in open-world detection and should not be claimed as a core contribution.

**Strengths Contributions:**

1.	This paper is well-written and easy-to-follow. Figures are clear to express the design and ideas.
2.	The essential factors in open-world perception of domain generalization and being open-ended are considered in this paper, which is crucial and lack investigation in previous works.
3.	The labeling pipeline for corner case scenarios and abnormal objects is designed, which is practical and useful.
4.	The vision-centric framework is proposed. The further fusion strategy is useful for enhancing accuracy while maintaining generalization ability.

---

> ### Author Rebuttal · Authors · 2025-07-31
>
> We thank the reviewer for recognition and comments of this work.
>
>
> **1. The proposed object detection framework has been investigated from previous 3D object detection approaches.**
>
> We propose a novel paradigm that is different from the previous 3D object detection paradigm. The previous paradigm extracts 3D features and then performs detection. Our paradigm leverages existing foundation models to generate 2D instance proposals, extracts instance-level features, and predicts 3D bounding boxes for each object. Only the converter needs to be trained. Our proposed converter is a network that is easy to train, easy to apply across datasets, and easy to augment.
>
> Furthermore, this paper's contributions extend beyond proposing an open-world detection method. They encompass our novel benchmark (including data and metrics), the annotation pipeline, the general-specialized integration concept, and the comprehensive analysis of existing approaches.
>
> **2. For the tasks of domain generalization and open-ended detection, the contribution of different methods to this issue will vary. Specify dataset construction and experiments are required to see what really matters for domain generalization or open-ended detection.**
>
> We contend that both domain generalization and open-world capability are fundamentally manifestations of a model's generalization capacity, which is influenced by architectural design and training data scale. Our paradigm is architecturally designed to ensure open-ended capability while leveraging foundation models trained on large-scale 2D data. To address the scarcity of real 3D data, we propose a converter that maximizes the utilization of diverse 3D data sources. Collectively, these strategies achieve more comprehensive data utilization.
>
> Section 3 of the paper details the dataset construction, while Section 6 presents the experimental details and analysis. Should you have any questions regarding the dataset creation or experimental setup, or if there are specific experiments you would like to see included, please kindly specify them. We are fully prepared to provide comprehensive explanations and will gladly revise the manuscript according to your feedback.
>
> **3. The ensemble of baselines naturally brings performance gains.**
>
> Section 5.2 focuses on General and Specialized Fusion, not mere ensemble methods. Ensemble techniques serve only as a preliminary means to implement General-Specialized Fusion. For comparison, we present in the table below results from: ensembles of two general models, and ensembles of two specialized models. Demonstrably, General-Specialized Fusion exhibits marked superiority over these normal ensembles. Most evidently, while normal ensembles employ NMS for deduplication, they struggle to achieve consistent AP improvement. In contrast, General-Specialized Fusion effortlessly enhances AP while delivering substantially greater gains in AR.
>
> | No. | Method | $AP$ | $AR$ | $AR^{nusc}_{seen}$ | $AR^{others}_{seen}$ | $AR^{nusc}_{unseen}$ | $AR^{others}_{unseen}$ |
> | --- | --- | --- | --- | --- | --- | --- | --- |
> | G1 | YoloWorld + Converter | 15.54 | 36.07 | 29.99 | 25.88 | 12.73 | 14.17 |
> | G2 | VL-SAM + Converter | 18.60 | 39.16 | 16.63 | 29.28 | 15.77 | 20.60 |
> | G1+G2 | OpenAD-Ens | 15.75 $\downarrow$ | 43.94$\uparrow$ | 33.81 | 34.86 | 21.51 | 24.61 |
> | S1 | BEVFormer | 14.43 | 22.73 | 51.86 | 16.59 | 0.00 | 0.03 |
> | S2 | BEVFusion | 15.57 | 33.50 | 59.93 | 20.64 | 0.00 | 0.00 |
> | S1+S2 | OpenAD-Ens | 14.54$\downarrow$ | 38.37$\uparrow$ | 64.29 | 27.37 | 0.00 | 0.03 |
> | G1+S2 | OpenAD-Ens | 16.30$\uparrow$ | 48.25$\uparrow$ | 64.84 | 39.11 | 10.59 | 16.85 |
> | G2+S1 | OpenAD-Ens | 18.74$\uparrow$ | 44.65$\uparrow$ | 57.54 | 35.68 | 15.77 | 20.63 |
> | G2+S2 | OpenAD-Ens | 18.90$\uparrow$ | 50.99$\uparrow$ | 65.10 | 41.86 | 15.77 | 20.60 |

---

> > ### Comment · Reviewer_QGow · 2025-08-04
> >
> > Thanks for the author's response. My concerns have been addressed mostly.  I choose to keep my positive score.

---

> > > ### Comment · Area_Chair_9BuC · 2025-08-07
> > >
> > > Given your "concerns have been addressed mostly", can you clarify what remaining concerns that are not addressed? This helps the authors to improve the paper further.

---

### Official Review · Reviewer_73ye · 2025-06-28

**Rating:** 5
**Confidence:** 4

**Summary:**

This paper introduces a new real open-world autonomous driving dataset and benchmark, covering multiple modalities (e.g.,camera, LiDAR) and providing a unified evaluation framework. The contribution lies in addressing the gap in existing ppen-world autonomous driving benchmarks, and the experiments validate its effectiveness. Overall, the paper is of high quality, though some details could be improved.

**Dataset Code Accessibility:**

Yes

**Ethical Considerations:**

No, there are no or only very minor ethics concerns

**Final Justification:**

Overall, this paper demonstrates excellence in innovation, experimental design, and theoretical analysis, addressing most of the core issues raised in the review process. The unresolved issues have minimal impact on the overall contribution, and therefore, I believe this paper meets the acceptance criteria. I recommend acceptance.

**Limitations Weaknesses:**

-Consider adding more multimodal tasks (e.g., occupancy) to further enhance the dataset’s utility.
-It is better to provide more multimodal demonstrations of corner cases to help readers better understand the capabilities of different sensors.
-The Partial Encoder and Depth Net mentioned in line 207 should be elaborated in the implementation details section, including specific reference literature.
-In the OpenAD baseline approach, the paper mentions that it is lightweight. Please provide the specific runtime and FLOPS.
-It is recommended to add corresponding captions to the two subplots in Figure 2 to explain the specific content of the diagrams. This would enhance the understanding of the Data composition of OpenAD. The current version is relatively cumbersome and makes it difficult to grasp the key points.

**Strengths Contributions:**

-The paper is clearly written, with detailed explanations of the dataset construction, annotation methods, and benchmark task definitions.
- Some figures provide concise illustrations. For example, Figure 3 is very informative and concise, and I could clearly get the overall idea at first glance.
-The overall structure of the paper is clear, and the language is fluent.

---

> ### Author Rebuttal · Authors · 2025-07-31
>
> We thank the reviewer for recognition and comments of this work.
>
>
> **1. Consider adding more multimodal tasks (e.g., occupancy) to further enhance the dataset’s utility.**
>
> In the future, we will incorporate evaluations for open-world occupancy prediction and instance segmentation, while expanding data to enhance scope and diversity.
>
> **2. It is better to provide more multimodal demonstrations of corner cases.**
>
> We will add LiDAR visualizations of corner cases and include corresponding textual discussions in the paper.
>
> **3. Partial Encoder and Depth Net**
>
> We employ Swin-tiny as the Partial Encoder and a multi-layer convolutional network as the Depth Net. The partial images have a resolution of 224*224. We will incorporate these details into the Implementation Details section.
>
> **4. The paper mentions that the proposed 2D-to-3D Converter is lightweight and easy to train.**
>
> Since the image resolution and point cloud input are dynamic, FLOPs are difficult to compute. We tested the training time on nuScenes and average inference time on the OpenAD data on a single NVIDIA A40 GPU.
>
> | Method | Training epochs on nuScenes | Training time on nuScenes |
> | --- | --- | --- |
> | BEVFormer | 24 epochs | 1~2 days on 8 A40 GPUs |
> | BEVStereo | 20 CBGS epochs (=90 epoches without CBGS) | 2~3 days on 8 A40 GPUs |
> | SparseBEV | 36 epochs | 3~4 days on 8 A800 GPUs |
> | **our 2D-to-3D converter** | **5 epochs (early stopping)** | **4~5 hours on 8 A40 GPUs** |
>
> For training, our proposed converter achieves significantly faster training speeds compared to existing 3D object detection models, while offering greater flexibility and enhanced cross-dataset and data-augmentation adaptability.
>
> For inference, the inference time of GenerateU is **1910ms** (on a single A40), and the inference time of our proposed converter is **184ms** (FP16, on a single A40), requiring only **9.6% additional inference time** to complete the conversion from 2D instances to 3D instances. We also have a lighter implementation: using a four-layer convolutional network as the Partial Encoder, the inference time of this model is **less than 20ms**, and its AR on OpenAD is 22.9.
>
> **5. It is recommended to add corresponding captions to the two subplots in Figure 2.**
>
> We will add subplot captions to Figure 2. Thank you for your suggestion.

---

### Official Review · Reviewer_XrvD · 2025-06-30

**Rating:** 5
**Confidence:** 4

**Summary:**

The paper presents OpenAD, the first benchmark for open-world autonomous driving 3D object detection. The dataset combines five public datasets (Waymo, Argoverse2, KITTI, nuScenes, and ONCE) with significantly different sensor setups and detection ranges. A multimodal large language-based method was developed to enable corner case discovery and semi-automatic annotation. As a result, 2000 scenarios were annotated with 2D and 3D bounding boxes using a unified format. Furthermore, a lightweight 2D-to-3D Bbox converter network was also trained as a vision-centric 3D open-world detector baseline. The benchmark results shed light on the possibility of ensembling general and specified models, effectively combining the benefits of open-world detectors and specialized (i.e., models trained on a closed label set) models.

The paper addresses two relevant problems:
1. Current 3D object detection models do not generalize well to new sensor setups,
2. Lack of 3D open-world datasets in autonomous driving.

Problem (1) is tackled by cross-dataset training described in the Appendix. The results show a large improvement over training only on one dataset (which is not surprising considering the sources of OpenAD). The creation of OpenAD is aimed at alleviating the issue raised by problem (2).

**Dataset Code Accessibility:**

Partly

**Dataset Code Comments:**

The code and installation/execution documentation are available on GitHub. Checkpoints or links to checkpoints are missing. Providing this information would increase the value of the submission and ease reproducibility.

**Ethical Considerations:**

No, there are no or only very minor ethics concerns

**Final Justification:**

The authors propose a valuable contribution to the community. The rebuttal addressed my concerns (as I see it, is consistently true for all reviewers); therefore, I increased my rate and propose to accept the paper.

**Limitations Weaknesses:**

Weaknesses:

- The dataset has only 2D/3D bounding boxes and misses for other tasks (e.g., segmentation), as it is acknowledged in the Limitations section.

The paper extends the evaluation metric "utilizing natural language annotations and conduct multi-threshold matching against string-formatted model predictions". This part needs clarification. Did the authors send through the class prediction on a CLIP model, and then compare the cosine similarity against the CLIP text embedding of the GT?

Questions:
- Did you verify the result of the MLLM Abnormal Filter for each frame of each scenario? How well does it work considering false negatives (i.e., not recognized hazardous objects)?
- The proposed baseline model heavily relies on depth estimation. Has the detection performance in distant areas (over 80 meters) been investigated? Distant detection is important on highways where the ego vehicle might need to stop from high speed when observing an undrivable surface caused by, e.g., obstacles fallen from trucks.

**Strengths Contributions:**

Strengths:

- The proposed dataset and benchmark are relevant and useful for evaluating domain adaptation and open-world detection capabilities of existing models
- Large-scale dataset & diverse scenarios with manually corrected annotations.

The evaluation compares general (open-world) and specialized (trained on a specific 3DOD dataset) detection models as well as the proposed baseline model with different backbones and using ensembling both in 2D and 3D detection settings. The results show that OpenAD-Ens method frequently overperforms the open-world and specialized models in almost all metrics. An interesting byproduct is the evaluation of MLLMs to test detection and semantic recognition capabilities mentioned in Section 4.

---

> ### Author Rebuttal · Authors · 2025-07-31
>
> We thank the reviewer for recognition and comments of this work.
>
>
> **1. The dataset has only 2D/3D bounding boxes and misses for other tasks.**
>
> In the future, we will incorporate evaluations for open-world occupancy prediction and instance segmentation, while expanding data to enhance scope and diversity.
>
> **2. Calculation of semantic similarity.**
>
> We encode the ground truth text and prediction text using the CLIP model's text encoder and compute the cosine similarity of their text features. For the specific code implementation, please refer to lines 203-219 in ```openad/evaluate/metrics_2d.py```. We will revise the corresponding text to enhance clarity.
>
> **3. How well does the MLLM Abnormal Filter work?**
>
> After examining scenes not filtered by the MLLM Abnormal Filter, we identified a notable number of scenes containing abnormal objects that went unreported. Therefore, we added the Manual Scene Filtering step.
>
> Moreover, the MLLM exhibits notable inconsistency—when presented with identical questions and images, it may alternately respond 'Yes' or 'No'. Therefore, our MLLM Abnormal Filter queries each scenario five times. If ≥2 responses are 'Yes', the scenario is provisionally flagged as abnormal. The ```VQA (Abnormal Yes/No, hit twice in 5 times)``` scores in the table below were obtained using this strategy.
>
> Furthermore, **OpenAD can also serve as an MLLM benchmark**. Besides directly requesting bounding boxes and calculating metrics, it can also be used as a Vision QA benchmark to inquire about the presence of abnormal objects or to request the specific semantics. The following table shows the test results for two open-source MLLMs. ```VQA (Abnormal Yes/No)``` simply requires the MLLM to determine whether an abnormal object exists. A "yes" answer is scored as 1, and a "no" answer is scored as 0. The average score is calculated, which can be used to measure false negatives. ```VQA (Semantic Description)``` requires the MLLM to describe objects in the scene that may affect traffic. Assuming there are $n$ categories, and the model predicts $m$ of them, it receives a score of $m/n$, and then calculates the average score. ```Detection (2D AR)``` is the detection metric in the paper.
>
> | Model | VQA (Abnormal Yes/No) | VQA (Abnormal Yes/No, hit twice in 5 times) | VQA (Semantic Description, hit once in 5 times) | Detection (2D AR, 5 times) |
> | --- | --- | --- | --- | --- |
> | ShareGPT4v 7B | 41.1% | 54.6% | 34.3% | 0.47% |
> | InternVL 2.5 8B | 46.9% | 64.2% | 39.4% | 0.84% |
>
> Experimental results demonstrate that even in binary-choice VQA tasks, the tested MLLMs exhibit a substantial number of False Negatives. When required to provide specific semantic descriptions, the recall rate drops to 30%-40%. Moreover, the MLLMs struggle to predict multiple bounding boxes directly.
>
> **4. Detection performance in distant areas.**
>
> | Distance (object number) | 0~10m (1890) | 10~40m (10192) | 40~80m (5544) | >80m (2135) | >80m（2135, relaxed thresholds）|
> | --- | --- | --- | --- | --- | --- |
> | BEVFormer | 18.89 | 24.06 | 7.22 | 0.00 | 0.00 |
> | SparseBEV(V2-99) | 14.50 | 16.03 | 5.56 | 0.00 | 0.00 |
> | BEVFormerv2 | 24.21 | 33.72 | 13.49 | 0.00 | 0.00 |
> | BEVFusion | 25.45 | 26.51 | 12.72 | 0.00 | 0.00 |
> | OpenAD-G | 45.58 | 34.21 | 12.61 | 0.40 | 7.50 |
> | OpenAD-Ens (OpenAD-G + BEVFusion) | 60.56 | 51.70 | 22.75 | 0.40 | 7.50 |
>
> 4.1 For Method
>
> We calculated AR for several baselines according to distance, as shown in the table above. Our baseline method achieves higher AR than other 3D Object Detection methods for distant detection. However, all models tested perform inadequately beyond 80 meters. While our model can provide some predictions, its positional error exceeds 2 meters. Other tested models, both depth-based (BEVFusion) and transformer-based (BEVFormer series, SparseBEV) architectures, fail to yield any predictions beyond 80m. These test results are entirely expected for the following reasons:
>
> - **The sensor configuration of publicly available data**: Public datasets mostly using 32-beam LiDAR, with virtually no point clouds beyond 80 meters. At the same time, the image resolution is low.
> - **The lack of training data**: All models, including our proposed converter, are trained on the nuScenes dataset, which contains almost no annotations farther than 80 meters.
> - **Inappropriate evaluation metrics**: For close objects, a prediction error of 1 meter is unacceptable, but for objects 100 meters away, a prediction error of 1 meter seems acceptable. This suggests that the currently commonly used fixed IoU or distance thresholds may be too strict for distant objects. In the above table, we also provide test results with the threshold relaxed to {4m, 6m, 8m, 10m}.
>
> Currently, the focus of this method is on open-ended and generalization capabilities. We adopted a dual-branch approach because we felt the pseudo-point cloud branch was heavily dependent on absolute depth, so we designed a convolutional branch to refine this. In future work, we will further improve the method.
>
> 4.2 For Benchmarking
>
> **OpenAD has 241 Expressway scenes and thousands of annotated distant objects, enabling distant area evaluation.**
>
> Detection for distant areas is indeed a challenging problem worthy of discussion. In future work, we will consider undertaking further research focused on distant area perception, such as finding higher-quality open-source data and developing specialized evaluation metrics.
>
>
> **5. Checkpoint Accessibility**
>
> We guarantee that all code and model checkpoints for the proposed baseline method will be made publicly available upon acceptance of this paper.

---

> > ### Comment · Reviewer_XrvD · 2025-08-01
> > **Reply to the rebuttal**
> >
> > Thank you to the authors for the detailed answer. My concerns have been addressed. I especially appreciate the investigation of detection performance in distant areas since this is an underexplored but important area. I agree with the author's arguments and find the relaxed evaluation in the far region a reasonable solution. Despite the challenges, their model performed better than baselines.

---

> > > ### Author Response · Authors · 2025-08-01
> > > **Thank you for the feedback on the rebuttal**
> > >
> > > We thank you for reading our rebuttal and confirming that all questions have been answered. As such, please consider raising the rating for this paper.
> > >
> > > Thank you,

---

> > > > ### Comment · Reviewer_XrvD · 2025-08-04
> > > >
> > > > I will increase my score. Thanks again for the detailed answers.

---

### Official Review · Reviewer_gikv · 2025-07-02

**Rating:** 5
**Confidence:** 5

**Summary:**

The paper introduces OpenAD, the first real-world open-world benchmark for 3D object detection in autonomous driving. It addresses three key challenges in open-world 3D perception: the lack of standardized benchmarks, limited training data, and low precision of open-world models. OpenAD integrates and re-annotates 2,000 scenes from five major driving datasets (e.g., KITTI, nuScenes, Waymo) using a pipeline powered by multimodal large language models (MLLMs) to identify and label 206 rare object categories. It proposes a vision-centric 3D object detection framework that lifts 2D open-world detections to 3D, and a fusion strategy to combine general open-world and specialized models. The authors also design novel metrics to evaluate semantic and spatial alignment for open-ended detection tasks. Experiments demonstrate improved generalization and performance over existing baselines using the new benchmark.

**Dataset Code Accessibility:**

Yes

**Ethical Considerations:**

No, there are no or only very minor ethics concerns

**Limitations Weaknesses:**

-  OpenAD currently supports only 2D and 3D object detection tasks and does not include other crucial open-world perception tasks such as occupancy prediction or instance segmentation

- Despite automation via MLLMs, the annotation process still relies heavily on manual filtering and correction, limiting scalability.

- Some baseline models show signs of overfitting on in-domain benchmarks like nuScenes, potentially skewing comparisons and requiring stronger cross-dataset validations.

**Strengths Contributions:**

- The paper introduces OpenAD, the first real-world benchmark specifically designed for evaluating 3D open-world object detection in autonomous driving.

- It contributes a unified annotation pipeline leveraging multimodal large language models (MLLMs) to identify and semantically label 206 rare object categories across five major datasets.

- The benchmark includes novel evaluation metrics that integrate both semantic and positional thresholds, enabling accurate assessment of open-ended predictions with synonym robustness.

- The authors propose a vision-centric baseline that lifts 2D open-world predictions to 3D using a lightweight 2D-to-3D converter, achieving strong generalization across domains.

- The paper further introduces a fusion method that combines open-world and specialized models, demonstrating complementary strengths and improved performance.

- Compared to prior datasets like CODA and AnoVox, OpenAD offers greater scale, diversity, and real-world complexity, filling a critical gap in the evaluation of open-world 3D perception.

- The paper is well-written and clearly structured, with informative figures and extensive experimental results that support the key claims.

---

> ### Author Rebuttal · Authors · 2025-07-31
>
> We thank the reviewer for recognition and comments of this work.
>
> **1. OpenAD currently supports only 2D and 3D object detection tasks.**
>
> In the future, we will incorporate evaluations for open-world occupancy prediction and instance segmentation, while expanding data to enhance scope and diversity.
>
> **2. The annotation process still relies heavily on manual filtering and correction.**
>
> Since OpenAD annotations are used in the benchmark, it is imperative to ensure their absolute accuracy, making manual correction at every stage indispensable. However, if the annotations are intended for training data where a certain error rate is tolerable, it may be feasible to eliminate some manual steps from the annotation pipeline.
>
> Furthermore, even with manual correction, the proposed semi-automated annotation pipeline can still save considerable time. In fully manual annotation scenarios, it is necessary to draw 3D bounding boxes from scratch and edit the semantic labels for each box, which typically takes approximately **120 seconds** per frame. With the proposed pipeline, only fine-tuning of position, size, and orientation angles is needed, with semantic edits required for only about 10% of the boxes. This approach reduces the annotation time per frame to **under 30 seconds**.
>
> **3. Some baselines show signs of overfitting on in-domain benchmarks like nuScenes.**
>
> As described in Lines 259-263, by calculating the in-domain AR and cross-domain AR ($AR^{nusc}$ and $AR^{others}$ in Table 3), we can determine whether a model overfits. For example, in the comparison of BEVStereo R50 and V2-99 in Table 3, although the in-domain AR improves with increasing backbone size, the out-domain AR decreases, and the overall AR also decreases. This is the contribution and advantage of our OpenAD benchmark.

---

> > ### Comment · Reviewer_gikv · 2025-08-02
> >
> > Thanks for the rebuttal, which addressed my concerns. I will keep my original rating as accept.

---

### Note · Authors · 2025-08-13

We sincerely thank the Area Chair and all reviewers for their thoughtful and constructive feedback on our work.
﻿
In the original submission, the reviewers all gave positive ratings and recognized several key strengths of our work:
﻿
1. **Novelty.** OpenAD is the first real-world open-world benchmark for 3D object detection in autonomous driving, introducing a unified annotation pipeline powered by multimodal large language models (MLLMs) and proposing a new open-ended 3D object detection framework.
2. **Importance.** The benchmark addresses critical gaps in open-world 3D perception, including the lack of standardized evaluation, limited open-ended perception data, and low precision of open-world models.
3. **Clarity.** The paper is well-structured, clearly written, and supported by informative figures and extensive experimental results.
4. **Effectiveness.** The proposed baseline and fusion strategy demonstrate improved generalization and performance over existing methods.
﻿
During the response and discussion process, we engaged in further discussions on the following aspects:
﻿
1. OpenAD can serve as a benchmark for MLLMs. We present evaluation results for two MLLMs. The experimental findings indicate that the tested MLLMs also struggle to solve open-ended object detection.
2. The OpenAD benchmark enables the evaluation of open-world detection for distant objects. We discuss the main challenges in distant object detection. The OpenAD method also demonstrates advantages over baseline methods on the OpenAD benchmark.
3. We provide further detailed information to illustrate the lightweight nature and ease of training of the proposed 2D-to-3D converter.
4. We compare against different model ensemble approaches to highlight the distinction between General-Specialized Fusion and conventional model ensemble.
﻿
We are pleased to see that all four reviewers have indicated that their main concerns have been addressed. We believe the revised work delivers a novel and impactful benchmark and an effective baseline method that will significantly advance research in open-world 3D perception for autonomous driving.

---

### Decision · Program_Chairs · 2025-09-18

**Decision:**

Accept (poster)

**Comment:**

**Paper summary.**
The paper introduces a new benchmark dataset called OpenAD to foster research of open-world autonomous driving through image-based 2D and 3D detection tasks. It collects publicly-available datasets (Argoverse 2, KITTI, nuScenes, ONCE, and Waymo) and re-annotates the images with respect to class names, seen/unseen tags during training, scenes, and 2D&3D boxes, and corner cases. In particular, it leverages MLLM to facilitate obstacle discovery and semantic labeling. It releases toolkit code as well. Further, the paper presents baseline methods for image-based 3D detection by learning to convert 2D box to 3D cuboid, and for general and specialized models fusion.


**Review summary.**
Four reviewers recommend to accept this paper after reading the rebuttal and acknowledge that the rebuttal has addressed their concerns. However, Reviewers gikv and 73ye provided some superficial comments and gave Accept ratings. Hence, their ratings are down-weighted. All the reviewers recognize the merits of the introduced new benchmark dataset to study 3D detection in the open world through autonomous driving, and the well-written paper. Below are specific points worth highlighting:

- Reviewers gikv, XrvD, and 73ye think it is a weakness for not including more tasks such as instance segmentation and occupancy prediction. However, the AC thinks it is fine as the paper positions itself as an open-world benchmark through the lens of 3D detection as stated in the paper title.

- The AC does not find the 206 class names in the main paper or supplementary document. Figure 1 and 3 contain "dog" as an example as a corner case and Figure 1 caption states `"These object categories have not been encountered by models trained on common 3D perception datasets during their training phase"`, but nuScenes does annotate an "animal" class. Therefore, the AC is confused what classes are novel. Are the so-called novel classes subclasses of a known class, e.g., dog (shown in paper) vs. animal (defined in nuScenes), scooter (listed in the suppl) vs. "Portable Personal Mobility Vehicle" (defined in nuScenes)? More fundamentally, how to understand "novel categories", either seen in the training set but unlabeled by the original dataset, or never-seen in any training data? The paper is expected to provide a more in-depth discussion.

- The AC cannot find important details in the paper: (1) Does OpenAD provide train/val/test splits? (2) What data is used as training set in OpenAD, especially given that OpenAD gathers multiple datasets? Reviewer XrvD digs into the code and finds specific lines to partially answer them. The AC strongly suggests writing clearly training protocols in the main paper so that readers can easily use the benchmark for future research.

**Decision.**
Based on the above, AC recommends accepting this paper but strongly encourages the authors to address the aforementioned points in the final version to make the benchmark more credible and beneficial to the community.